# Strategy-Aware Contextual Bandits

Keegan Harris[1], Chara Podimata[2,3], and Zhiwei Steven Wu[1]

[1]Carnegie Mellon University
[2]MIT
[3]University of California, Berkeley
{keeganh,zstevenwu}@cmu.edu, podimata@mit.edu

## Abstract

Algorithmic tools are often used to make decisions about people in high-stakes domains. In the presence of such automated decision making, there is incentive for strategic agents to modify their input to the algorithm in order to receive a more desirable outcome. While previous work on *strategic classification* attempts to capture this phenomenon, these models fail to take into account the *multiple actions* a decision maker usually has at their disposal, and the fact that they often have access only to *bandit feedback*. Indeed, in standard strategic classification, the decision maker's action is to either assign a positive or a negative prediction to the agent, and they are assumed to have access to the agent's true label after the fact. In contrast, we study a setting where the decision maker has access to *multiple actions* but only can see the outcome of the action they assign. We formalize this setting as a *contextual bandit* problem, in which a decision maker must take actions based on a sequence of strategically modified contexts. We provide an algorithm with no regret compared to the best fixed policy in hindsight *if the agents' were truthful when revealing their contexts* (i.e., no-strategic-regret) for the two action setting, and prove that sublinear strategic regret is generally not possible for settings in which the number of actions is greater than two. Along the way, we obtain impossibility results for multi-class strategic classification which may be of independent interest.

## 1 Introduction

When subjugated to algorithmic decision making, decision subjects may strategically modify their input to the algorithm in order to receive a better outcome.This may be viewed as a form of *distribution shift* in which the data distribution the machine learning (ML) algorithm observes has been strategically altered. It is therefore desirable to design reliable ML algorithms in the presence of such *strategic interactions*. While this area of study has received much recent interest in the literature under the name of *strategic classification* (see Section 2 for a list of related work), current work in this area fails to capture several salient features of reality, as most models only consider *binary classification* or *linear regression* settings under *full feedback* (i.e., the decision maker sees the agent's true label after making a prediction). In contrast, we study a model in which a decision maker must assign a strategic agent one of *several actions* under *bandit feedback* (i.e., the decision maker only observes some *reward* for the action they assign). Our model captures several settings of practical interest including (1) personalized shopping, in which an online platform (decision maker) must decide which level of discount (action) to give to give to customers (strategic agents) given their previous buying patterns and (2) lending, in which a bank (decision maker) needs to decide which type of loan (e.g., high-interest, low-interest, no loan) to offer to applicants (strategic agents) given their credit history. Our main contributions are as follows: (1) We formalize the *strategy-aware contextual bandit setting*, in which a decision maker must assign a sequence of strategic agents one of $n$ actions, and observes their outcomes under bandit feedback. (2) For $n = 2$ actions, we

2022 Trustworthy and Socially Responsible Machine Learning (TSRML 2022) co-located with NeurIPS 2022.

---

**Interaction protocol: Contextual bandits under strategic manipulations**

For each round $t \in [T]$:

1. Decision maker publicly commits to a mapping $\pi_t : \mathcal{X} \to \mathcal{A}$. Agent $\Gamma_t$ arrives with private type $\mathbf{x}_t$.

2. Agent $t$ strategically selects context $\mathbf{x}'_t$ according to Assumption 3.3.

3. Decision maker observes context $\mathbf{x}'_t$ and plays action $a_t = \pi_t(\mathbf{x}'_t)$.

4. Decision maker receives reward $r_t^D(a_t)$ and agent receives reward $r^A(a_t)$.

---

Figure 1: Description of the strategy-aware contextual bandit setting we consider, which may be viewed as a generalization of the contextual bandit problem with linear rewards.

show that it is possible to minimize *strategic regret*, a strong notion of hindsight rationality. (3) Finally, we show that in sharp contrast to the two-action setting, when $n \geq 3$ there exist cases where achieving sublinear strategic regret is not possible. Along the way, we obtain impossibility results for multi-class strategic classification which may be of independent interest. In particular, we show that there exist situations in which perfect classification under strategic responses is not possible with three or more labels, even if agents are linearly separable before strategic modification.

## 2 Related Work

Our work is closely related to two lines of work: *learning in the presence of strategic behavior* (e.g., [3, 5, 9, 12, 14, 15, 18, 13, 6, 11, 17, 19, 7]) and *contextual bandits* (see e.g. [16, 21, 22, 20, 8, 1, 10, 2] for a highly incomplete list).

**Strategic learning.** The problem of *strategic classification* was first introduced in Hardt et al. [12]. The original formulation considers a sequential game between a "jury" (decision maker), who publishes a classifier, and a "contestant" (strategic agent), who best responds by strategically modifying their observable features. In the years since, there have been extensions to online learning (Dong et al. [9], Chen et al. [7], Ahmadi et al. [3]), repeated interactions (Harris et al. [14]), and social learning settings (Bechavod et al. [5]), although to the best of our knowledge, we are the first to consider multi-class strategic classification under bandit feedback.

**Contextual bandits.** The contextual bandit setting can be thought of a multi-armed bandit problem, in which the bandit algorithm has access to extra information about the reward of each action (i.e., the *context*) at every round. Various relationships between contexts and rewards have been studied; several of the most common are *Lipschitz contextual bandits* (e.g., Hazan and Megiddo [16], Lu et al. [21], Slivkins [22]), in which the expected reward of each arm is Lipschitz in the context, *linear contextual bandits* (e.g., Li et al. [20], Chu et al. [8]), where the expected reward is linear in the context, and *agnostic algorithms* (e.g., Agarwal et al. [2], Foster and Rakhlin [10]), which make no assumptions on the underlying reward distribution, but assume access to various classification/regression oracles. In particular, our setting is most similar to that of linear contextual bandits, although with the notable exception that in our setting, the expected reward of each action is linear in the agent's *private type*. Since each agent's private type is not observed by the decision maker, this introduces significant additional difficulties in the decision maker's learning problem and leads to our impossibility results in Section 5.

## 3 Model and Preliminaries

We consider a setting in which a *decision maker* interacts with a sequence of $T$ strategic *agents*. At each time $t \in [T]$, a new agent $\Gamma_t$ arrives with a *private type* $\mathbf{x}_t \in \mathcal{X} \subset [-L, L]^d$ for $L \in \mathbb{R}_+$, and presents a *context* $\mathbf{x}'_t \in \mathbb{R}^d$ to the decision maker. Note that it is possible for $\mathbf{x}'_t = \mathbf{x}_t$, but the two do not necessarily coincide. Given a context $\mathbf{x}'_t$, the decision maker takes an *action* $a_t$ from some set of actions $\mathcal{A}$, where $\mathcal{A} = \{1, \dots, n\}$. For each action $a \in \mathcal{A}$, we assume that the decision maker's

*expected reward* $\mathbb{E}[r_t^D(a)] \in [-1, 1]$ is a linear function of the agent's private type. In particular, we make the following assumption on the decision maker's reward.

**Assumption 3.1** (Linearity of Rewards). *The decision maker reward of each arm* $a \in \mathcal{A}$ *at time* $t$ *is given by*

$$r_t^D(a) = \langle \boldsymbol{\theta}_a, \mathbf{x}_t \rangle + \varepsilon_t$$

*for some* $\boldsymbol{\theta}_a \in \mathbb{R}^d$ *which is unknown to the decision maker, where* $\varepsilon_t$ *is zero-mean sub-Gaussian noise.*

Note that if $\mathbf{x}_t' = \mathbf{x}_t \ \forall t \in [T]$, our setting reduces to that of linear contextual bandits (e.g, [20, 2]). In contrast to the decision maker's reward, we assume that each agent's reward $r_t^A(a)$ is a function of the action alone, i.e., $r_t^A(a) = r^A(a), \forall t \in [T]$, and is known to the decision maker. In the personalized shopping example, a consumer's private type can be thought of as the type of customer they are (e.g., loyal customer, new customer, thrifty customer, etc.). If customers knew the online platform was going to offer them a discount based on their buying habits, they may strategically alter their short-term behavior on the platform (i.e., their context) in order to obtain the highest discount possible.

The decision maker's *policy* $\pi_t : \mathcal{X} \to \mathcal{A}$ at time $t$ is a mapping from agent contexts to actions. In particular, the goal of the decision maker is to deploy a sequence of policies $\pi_1, \ldots \pi_T$ in order to minimize *strategic regret*, which is defined as follows:

**Definition 3.2** (Strategic Regret). *The expected strategic regret of a sequence of policies* $\{\pi_t\}_{t=1}^T$ *is defined as the cumulative expected reward of* $\pi^*(\mathbf{x}) := \arg\max_{a \in \mathcal{A}} \sum_{t \in [T]} \mathbb{E}[r_t^D(a)|\mathbf{x}]$, *the optimum-in-hindsight policy given the agents' private types, minus the cumulative expected reward of the deployed sequence of policies. Formally,*

$$\mathbb{E}[R(T)] = \sum_{t=1}^T \underbrace{\mathbb{E}[r_t^D(\pi^*(\mathbf{x}_t))|\Gamma_t]}_{\text{reward of optimal policy given private type}} - \underbrace{\mathbb{E}[r_t^D(\pi_t(\mathbf{x}_t'))|\Gamma_t]}_{\text{realized reward given context}}$$

A decision maker with sublinear strategic regret will approach the performance of the best-in-hindsight policy if agents are *not* strategic as $T \to \infty$, thereby learning to "account" for the strategic behavior of agents in the long run. Given the decision maker policy $\pi_t$, it is natural for an agent to choose their context in a way which maximizes their reward. Specifically, we assume that agent $\Gamma_t$ determines their context by strategically modifying their private type based on the decision maker's policy, subject to a constraint on the amount of modification which is possible. In addition to being a common assumption in the literature (e.g., [7, 18, 14]), this *budget* constraint on the amount an agent can modify their private type reflects the fact that agents have inherent constraints on the amount of time and resources they can spend on modification.

**Assumption 3.3** (Agent Best Response). *We assume that agent* $\Gamma_t$ *best-responds to the decision maker's policy* $\pi_t$ *in order to maximize their expected reward, subject to the constraint that their context* $\mathbf{x}_t'$ *is within a* $\delta$-radius of their private type $\mathbf{x}_t$. *Formally, we assume that the agent solves the following optimization to determine their context:*

$$\mathbf{x}_t' \in \arg\max_{\mathbf{x}' \in \mathcal{X}} \ r^A(\pi_t(\mathbf{x}'))$$
$$s.t. \ \|\mathbf{x}' - \mathbf{x}_t\|_2 \leq \delta$$

Furthermore, we assume that if an agent is indifferent between modifying their context and not modifying, they choose not to modify. See Figure 1 for a summary of the setting we consider. Finally, we conclude this section by introducing the idea of an *equivalence region*, which will be useful when discussing our theoretical contributions. Roughly speaking, an equivalence region corresponds to all contexts for which a policy $\pi(\cdot)$ assigns action $a$.

**Definition 3.4** (Equivalence region). *An equivalence region for an action* $a \in \mathcal{A}$ *under policy* $\pi$ *is defined as the set of all* $\mathbf{x} \in \mathcal{X}$ *such that* $\pi(\mathbf{x}) = a$.

For the policies we consider, each equivalence region will be a closed, convex set.

## 4  Sublinear strategic regret for two actions

We now outline an algorithm which achieves sublinear strategic regret in the two-action setting. W.l.o.g., we assume that $r^A(2) > r^A(1)$ for the remainder of the subsection. While our algorithm

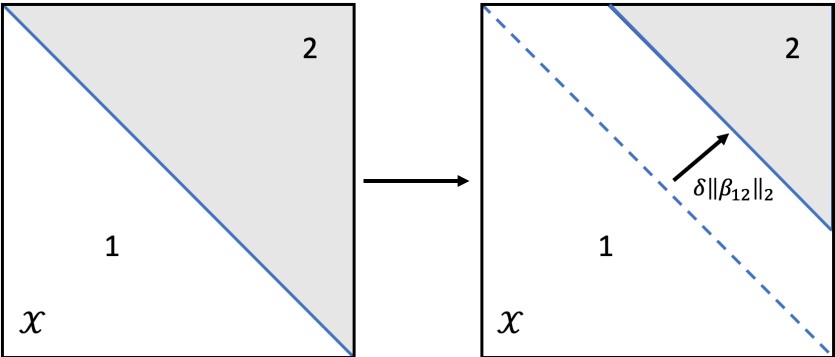

Figure 2: (Left: Proposition 4.1) The optimal policy in the non-strategic setting assigns action 1 to the left of the line $\langle \boldsymbol{\beta}_{12}, \mathbf{x} \rangle = 0$ (blue) and action 2 to the right. (Right: Proposition 4.2) In the strategic setting, the optimal decision boundary is shifted by $\delta \|\boldsymbol{\beta}_{12}\|_2$. In both figures, the equivalence region for $a = 1$ is denoted by white and that for $a = 2$ in gray.

is similar to previous algorithms from the "strategic classification" literature (e.g., [12, 3]), we view it as an important structural result which (1) illustrates that sublinear strategic regret *is* possible in the two action setting and (2) provides intuition as to why this is generally not possible in the case of 3+ actions. We begin with a result for the offline, non-strategic setting.

**Proposition 4.1.** *In the two-action setting in which agents are not strategic (i.e., $\mathbf{x}'_t = \mathbf{x}_t, \forall t \in [T]$) and $\boldsymbol{\theta}_1, \boldsymbol{\theta}_2$ are known to the decision maker, the optimal policy assigns action $a_t = 1$ if $\langle \boldsymbol{\beta}_{12}, \mathbf{x}'_t \rangle \geq 0$ and action $a_t = 2$ otherwise, where $\boldsymbol{\beta}_{12} := \boldsymbol{\theta}_1 - \boldsymbol{\theta}_2$.*

Proposition 4.1 follows from Assumption 3.1. While the decision boundary of Proposition 4.1 is optimal in the non-strategic setting, its performance may suffer in the strategic setting (i.e., if $\mathbf{x}'_t \neq \mathbf{x}_t$). In order to account for potential strategic manipulations, the optimal decision boundary needs to be *shifted* by an amount proportional to the agents' ability to manipulate their private type. Intuitively, this shift disincentives strategic manipulations from the agents who would have manipulated without such shift.

**Proposition 4.2.** *In the two-action setting in which agents strategically reveal their contexts according to Assumption 3.3, the optimal policy assigns action $a_t = 1$ if $\langle \boldsymbol{\beta}_{12}, \mathbf{x}'_t \rangle \geq -\delta \|\boldsymbol{\beta}_{12}\|_2$ and action $a_t = 2$ otherwise, where $\boldsymbol{\beta}_{12} := \boldsymbol{\theta}_1 - \boldsymbol{\theta}_2$. Moreover, the performance of such a policy when agents strategically reveal their contexts according to Assumption 3.3 matches that of Proposition 4.1 in the non-strategic setting (i.e., when $\mathbf{x}'_t = \mathbf{x}_t, \forall t \in [T]$).*

See Figure 2 for a visual depiction of the policies of Propositions 4.1 and 4.2. Given the result of Proposition 4.2, it is rather straightforward to achieve sublinear strategic regret in the online setting by using the standard "explore-first" technique. In particular, under minor distributional assumptions on the population of agents (e.g., the covariance of the agent population has full rank), the decision maker can apply Algorithm 1 to achieve $\mathcal{O}(T^{2/3})$ strategic regret.

**Theorem 4.3** (Abridged; see Appendix A.2 for formal statement)**.** *For sufficiently large $T$, if $T_0 = \tilde{\mathcal{O}}(T^{2/3})$*

$$\mathbb{E}[R(T)] = \tilde{\mathcal{O}}(d^{1/3}T^{2/3}),$$

*where $\tilde{\mathcal{O}}(\cdot)$ suppresses terms which do not depend on $T, d$, as well as logarithmic terms.*

The idea of Algorithm 1 is as follows: the first $T_0$ rounds are the *exploration* phase, i.e., contexts are not used to make decisions about which action to assign. Since the agents have no incentive to modify their private type, $\mathbf{x}'_t = \mathbf{x}_t, \forall t \in [T_0]$. The decision maker then uses data from the explore phase to estimate $\boldsymbol{\beta}_{12}$, and acts greedily according to $\hat{\boldsymbol{\beta}}_{12}$ for the remaining $T - T_0$ rounds.

## 5 $\Omega(T)$ **strategic regret for more than two actions**

In the previous subsection, we showed that it is generally possible to achieve sublinear strategic regret in the two-action strategy-aware contextual bandit setting using a linear separating hyperplane, shifted

**ALGORITHM 1:** Explore-first for online strategic classification with bandit feedback (`Explore-First`)

---

**Data:** $n \geq 0$
**for** $t = 1, \ldots, T_0$ **do**
    **if** $t \leq T_0/2$ **then**
        Assign action $a_1$
    **else**
        Assign action $a_2$
    **end**
**end**
Estimate $\boldsymbol{\theta}_1, \boldsymbol{\theta}_2$ via OLS as $\hat{\boldsymbol{\theta}}_1, \hat{\boldsymbol{\theta}}_2$. Let $\hat{\boldsymbol{\beta}}_{12} := \hat{\boldsymbol{\theta}}_1 - \hat{\boldsymbol{\theta}}_2$.
**for** $t = T_0 + 1, \ldots, T$ **do**
    Assign action

$$a_t = \begin{cases} 1 & \text{if } \langle \hat{\boldsymbol{\beta}}_{12}, \mathbf{x}_t \rangle - \delta \|\hat{\boldsymbol{\beta}}_{12}\|_2 \geq 0 \\ 2 & \text{o.w.} \end{cases}$$

**end**

---

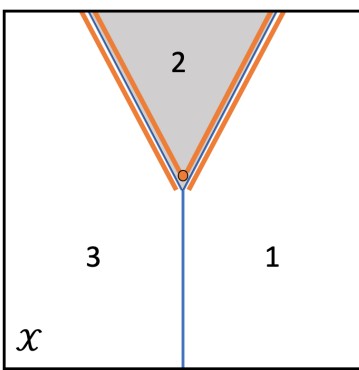

Figure 3: Visualization of the "bad example" from Theorem 5.1. Blue lines represent optimal decision boundaries in the non-strategic setting. The distribution over agent private types for which perfect classification is not possible puts probability mass $1/5$ on the orange dot and each of the orange lines.

by an appropriate amount. However, we will now show that this intuition does not carry over to the 3+ action setting. Furthermore, we show that there exist situations for which *no linear policy* (e.g., set of linear hyperplanes) can achieve sublinear strategic regret if the decision maker has three or more actions.[1] As was the case in Section 4, we begin with a result for the offline setting. Note that this result immediately implies that similar "bad examples" exist for settings in which $n > 3$ and $d > 2$.

**Theorem 5.1.** *Suppose* $\mathcal{X} = \mathbb{R}^2$, $n = 3$, *and* $r^A(2) > r^A(1) = r^A(3)$. *For* $\boldsymbol{\theta}_1 = [1 \quad 0.5]^\top$, $\boldsymbol{\theta}_2 = [0 \quad 1]^\top$, $\boldsymbol{\theta}_3 = [-1 \quad 0.5]^\top$, *there exists a distribution over agent private types such that no linear policy can achieve perfect classification if agents choose their contexts according to Assumption 3.3.*

*Proof Sketch.* The proof of Theorem 5.1 proceeds by constructing a "bad" distribution over agents (shown in orange in Figure 3) in which it is *necessary* to shift the separating hyperplanes $\boldsymbol{\beta}_{12}$ and $\boldsymbol{\beta}_{23}$ (defined analogously to $\boldsymbol{\beta}_{12}$ in Proposition 4.2) in order to achieve perfect classification on a certain subpopulation of agents (far-left and far-right orange lines in Figure 3). Next, we show that shifting $\boldsymbol{\beta}_{12}$ and $\boldsymbol{\beta}_{23}$ by such an amount will make it *impossible* to classify another agent subpopulation (orange dot in Figure 3). Finally, we conclude that perfect classification of this distribution of strategic agents is not possible. Note that this is in contrast to the non-strategic setting, in which this agent population is trivially linearly separable.

*Proof.* Recall that assigning $a_t = \arg\max_{a \in \mathcal{A}} \langle \boldsymbol{\theta}_a, \mathbf{x}_t \rangle$ will achieve perfect classification in the non-strategic setting. Define $\boldsymbol{\beta}_{12} := \boldsymbol{\theta}_1 - \boldsymbol{\theta}_2 = [1 \quad -0.5]^\top$, $\boldsymbol{\beta}_{13} := \boldsymbol{\theta}_1 - \boldsymbol{\theta}_3 = [2 \quad 0]^\top$, and

---

[1]We hypothesize that this impossibility holds for more expressive policy classes, and we view extending our results to such settings as an important direction for future research.

$\boldsymbol{\beta}_{23} := \boldsymbol{\theta}_2 - \boldsymbol{\theta}_3 = [1 \ 0.5]^\top$ Assigning actions according to the following set of decision boundaries is equivalent to assigning $a_t = \arg\max_{a \in \mathcal{A}} \langle \boldsymbol{\theta}_a, \mathbf{x}_t \rangle$:

$$a_t = \begin{cases} 1 & \text{if } \langle \boldsymbol{\beta}_{12}, \mathbf{x}_t \rangle \geq 0 \text{ and } \langle \boldsymbol{\beta}_{13}, \mathbf{x}_t \rangle \geq 0 \\ 2 & \text{if } \langle \boldsymbol{\beta}_{12}, \mathbf{x}_t \rangle < 0 \text{ and } \langle \boldsymbol{\beta}_{23}, \mathbf{x}_t \rangle \geq 0 \\ 3 & \text{if } \langle \boldsymbol{\beta}_{13}, \mathbf{x}_t \rangle < 0 \text{ and } \langle \boldsymbol{\beta}_{23}, \mathbf{x}_t \rangle < 0 \end{cases}$$

See Figure 3 for a visual depiction of the decision space (blue lines).

Consider the following agent private types: $\mathbf{x}_1 = [0 \ \alpha_1]^\top$, $\mathcal{X}_2 = \{\mathbf{x} : \langle \boldsymbol{\beta}_{12}, \mathbf{x} \rangle = \alpha_2, \ \mathbf{x}[1] > 0\}$, $\mathcal{X}_2' = \{\mathbf{x} : \langle \boldsymbol{\beta}_{12}, \mathbf{x} \rangle = \boldsymbol{\beta}_{12}[2] \cdot \alpha_1, \ \mathbf{x}[1] > 0\}$, $\mathcal{X}_3 = \{\mathbf{x} : \langle \boldsymbol{\beta}_{23}, \mathbf{x} \rangle = -\alpha_2, \ \mathbf{x}[1] < 0\}$, $\mathcal{X}_3' = \{\mathbf{x} : \langle \boldsymbol{\beta}_{23}, \mathbf{x} \rangle = \boldsymbol{\beta}_{23}[2] \cdot \alpha_1, \ \mathbf{x}[1] < 0\}$, where $\alpha_1, \alpha_2 > 0$ and the corresponding distribution over $\mathcal{X}$: $\mathbb{P}(\mathbf{x}_t = \mathbf{x}_1) = \frac{1}{5}$, $\mathbb{P}(\mathbf{x}_t \in \mathcal{X}_2) = \frac{1}{5}$, $\mathbb{P}(\mathbf{x}_t \in \mathcal{X}_2') = \frac{1}{5}$, $\mathbb{P}(\mathbf{x}_t \in \mathcal{X}_3) = \frac{1}{5}$, $\mathbb{P}(\mathbf{x}_t \in \mathcal{X}_3') = \frac{1}{5}$. Suppose that $r^A(2) > r^A(1) = r^A(3)$. Under this setting, we make use of the following two lemmas to identify necessary conditions on the hyperplane shift $\varepsilon$ in order to achieve perfect classification in the strategic setting:

**Lemma 5.2.** *To prevent gaming, the decision maker must assign action $a_2$ only if $\langle \boldsymbol{\beta}_{12}, \mathbf{x}_t \rangle + \varepsilon_{12} < 0$ and $\langle \boldsymbol{\beta}_{23}, \mathbf{x}_t \rangle - \varepsilon_{23} \geq 0$, where $\varepsilon_{12} \geq \delta \|\boldsymbol{\beta}_{12}\|_2 - \alpha_2$ and $\varepsilon_{23} > \delta \|\boldsymbol{\beta}_{23}\|_2 - \alpha_2$.*

**Lemma 5.3.** *To correctly classify agents with $\mathbf{x}_t = \mathbf{x}_1$, the decision maker must assign action $a_2$ only if $\langle \boldsymbol{\beta}_{12}, \mathbf{x}_t \rangle + \varepsilon_{12} < 0$ and $\langle \boldsymbol{\beta}_{23}, \mathbf{x}_t \rangle - \varepsilon_{23} \geq 0$, where $\varepsilon_{12} \leq -\boldsymbol{\beta}_{12}[2](\alpha_1 + \delta)$ and $\varepsilon_{23} \leq \boldsymbol{\beta}_{23}[2](\alpha_1 + \delta)$.*

An immediate consequence of the above two lemmas is that unless $\delta \|\boldsymbol{\beta}_{12}\|_2 - \alpha_2 \leq -\boldsymbol{\beta}_{12}[2](\alpha_1 + \delta)$ and $\delta \|\boldsymbol{\beta}_{23}\|_2 - \alpha_2 \leq \boldsymbol{\beta}_{23}[2](\alpha_1 + \delta)$, perfect classification is not possible. In the above setting, this implies that perfect classification is not possible unless $\frac{1}{2}\alpha_1 + \alpha_2 \geq \delta(\sqrt{1.25} - 0.5)$, which does not hold for sufficiently small $\alpha_1, \alpha_2$. $\qquad\square$

The following corollary follows immediately from Theorem 5.1 and Definition 3.2.

**Corollary 5.4.** *In the setting of Theorem 5.1, no online learning algorithm can achieve sublinear strategic regret.*

## 6 Conclusions and future research

In this extended abstract, we showed a separation between strategic classification and multi-class strategic classification, which immediately implies an impossibility result for the strategy-aware contextual bandit setting with three or more actions. There are several exciting directions for future research.

The most important direction for future work is to identify a set of *sufficient* or *necessary* conditions for which the impossibility result of Theorem 5.1 holds. In settings for which it does not hold, it would be interesting to see if ideas similar to the explore-exploit algorithm of Section 4 achieve sublinear strategic regret for 3+ actions. Another exciting direction would be to give a lower bound on strategic regret in the two-action setting. In particular, it is unclear if the $\Omega(T^{1/2})$ lower bound from the non-strategic setting applies, or if $\Omega(T^{2/3})$ strategic regret is the best one can do. If the $\Omega(T^{1/2})$ lower bound does indeed hold, deriving an algorithm which obtains this rate would be interesting. Finally, one may argue that strategic regret is too strong of a benchmark given the negative result of Theorem 5.1. Therefore, it may be desirable to derive algorithms with low *Stackelberg regret* [4] in the strategy-aware contextual bandit setting.

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

# A  Proofs from Section 4

## A.1  Proof of Proposition 4.2

**Proposition A.1.** *In the two-action setting in which agents strategically reveal their contexts according to Assumption 3.3, the optimal policy assigns action $a_t = 1$ if $\langle \boldsymbol{\beta}_{12}, \mathbf{x}_t \rangle \geq -\delta \|\boldsymbol{\beta}_{12}\|_2$ and action $a_t = 2$ otherwise, where $\boldsymbol{\beta}_{12} := \boldsymbol{\theta}_1 - \boldsymbol{\theta}_2$.*

*Proof.* Denote the unit vector along the direction of $\boldsymbol{\beta}_{12}$ as $\mathbf{n} = \frac{\boldsymbol{\beta}_{12}}{\|\boldsymbol{\beta}_{12}\|_2}$. We proceed on a case-by-case basis.

**Case 1**: The policy of Proposition 4.1 assigns action $a_t = 1$ to agent $\Gamma_t$:

Since $a_t = 1$, $\langle \boldsymbol{\beta}_{12}, \mathbf{x}_t \rangle \geq 0$. In order to move across the boundary under as little modification as possible, the agent should modify their context in the direction of $-\mathbf{n}$. However, the most they can manipulate their context by is $\delta$, and $\langle \boldsymbol{\beta}_{12}, \mathbf{x}_t - \delta \mathbf{n} \rangle \geq -\delta \|\boldsymbol{\beta}_{12}\|_2$ since $\langle \boldsymbol{\beta}_{12}, \mathbf{x}_t \rangle \geq 0$. Therefore no valid manipulations exist for which the policy of Proposition 4.2 assigns action $a = 2$ to agent $\Gamma_t$.

**Case 2:** The policy of Proposition 4.1 assigns action $a_t = 2$ to agent $\Gamma_t$:

Since $a_t = 2$, $\langle \boldsymbol{\beta}_{12}, \mathbf{x}_t \rangle < 0$. If agent $\Gamma_t$ modifies their private from $\mathbf{x}_t$ to context $\mathbf{x}_t - \delta \mathbf{n}$, $\langle \boldsymbol{\beta}_{12}, \mathbf{x}_t' \rangle = \langle \boldsymbol{\theta}_1 - \boldsymbol{\theta}_2, \mathbf{x}_t \rangle - \delta \langle \boldsymbol{\theta}_1 - \boldsymbol{\theta}_2, \mathbf{n} \rangle < -\delta \|\boldsymbol{\beta}_{12}\|_2$. Therefore, they can receive action $a = 2$ under the policy of Proposition 4.2 by strategically modifying their private type from $\mathbf{x}_t$ to context $\mathbf{x}_t - \delta \mathbf{n}$.

$\square$

## A.2  Proof of sublinear strategic regret for two actions

**Lemma A.2.** *If $T_0 \geq \frac{4dL}{\sigma_x^2 \log \frac{1}{2} e} \log \frac{2d}{\delta}$,*

$$|\langle \hat{\boldsymbol{\theta}}_1 - \boldsymbol{\theta}_1, \mathbf{x} \rangle| \leq \frac{4 L \sigma_\varepsilon \sqrt{d \log(2d/\delta)}}{\sqrt{T_0} \sigma_x^2}$$

*with probability at least $1 - \delta$. An identical bound holds for $|\langle \hat{\boldsymbol{\theta}}_2 - \boldsymbol{\theta}_2, \mathbf{x} \rangle|$.*

*Proof.*

$$
\begin{aligned}
\left| \langle \hat{\boldsymbol{\theta}}_1 - \boldsymbol{\theta}_1, \mathbf{x} \rangle \right| &\leq \|\hat{\boldsymbol{\theta}}_1 - \boldsymbol{\theta}_1\|_2 \|\mathbf{x}\|_2 \\
&\leq L\sqrt{d} \|\hat{\boldsymbol{\theta}}_1 - \boldsymbol{\theta}_1\|_2 \\
&= L\sqrt{d} \left\| \left( \sum_{t=1}^{T_0/2} \mathbf{x}_t \mathbf{x}_t^\top \right)^{-1} \sum_{t=1}^{T_0/2} \mathbf{x}_t (\mathbf{x}_t^\top \boldsymbol{\theta}_1 + \varepsilon_t) - \boldsymbol{\theta}_1 \right\|_2 \\
&= L\sqrt{d} \left\| \left( \sum_{t=1}^{T_0/2} \mathbf{x}_t \mathbf{x}_t^\top \right)^{-1} \sum_{t=1}^{T_0/2} \mathbf{x}_t \varepsilon_t \right\|_2 \\
&\leq L\sqrt{d} \left\| \left( \sum_{t=1}^{T_0/2} \mathbf{x}_t \mathbf{x}_t^\top \right)^{-1} \right\|_2 \left\| \sum_{t=1}^{T_0/2} \mathbf{x}_t \varepsilon_t \right\|_2 \\
&= \frac{L\sqrt{d} \left\| \sum_{t=1}^{T_0/2} \mathbf{x}_t \varepsilon_t \right\|_2}{\sigma_{min}(\sum_{t=1}^{T_0/2} \mathbf{x}_t \mathbf{x}_t^\top)} \\
&= \frac{L\sqrt{d} \left\| \sum_{t=1}^{T_0/2} \mathbf{x}_t \varepsilon_t \right\|_2}{\lambda_{min}(\sum_{t=1}^{T_0/2} \mathbf{x}_t \mathbf{x}_t^\top)}
\end{aligned}
$$

**Bounding** $\left\|\sum_{t=1}^{T_0/2} \mathbf{x}_t \varepsilon_t\right\|_2$

Since $\mathbf{x}_t(j) \in [-L, L]$ and $\varepsilon_t$ is a sub-Guassian random variable with variance $\sigma_\varepsilon^2$, $\mathbf{x}_t(j)\varepsilon_t$ is a zero-mean sub-Gaussian random variable with variance at most $L^2\sigma_\varepsilon^2$. Therefore, we can use the following result to bound $\left\|\sum_{t=1}^{T_0/2} \mathbf{x}_t \varepsilon_t\right\|_2$ with high probability.

**Theorem A.3** (High probability bound on the sum of unbounded sub-Gaussian random variables).
*Let $x_t \sim subG(0, \sigma^2)$. For any $\delta \in (0, 1)$, with probability at least $1 - \delta$,*

$$\left|\sum_{t=1}^{T} x_t\right| \leq \sigma\sqrt{2T\log(1/\delta)}.$$

Using Theorem A.3,

$$\left\|\sum_{t=1}^{T_0/2} \mathbf{x}_t \varepsilon_t\right\|_2 = \sqrt{\sum_{j=1}^{d}\left(\sum_{t=1}^{T_0/2} \mathbf{x}_t(j)\varepsilon_t\right)^2}$$

$$\leq \sqrt{\sum_{j=1}^{d}\left(L\sigma_\varepsilon\sqrt{T_0\log(1/\delta_j)}\right)^2}$$

$$= L\sigma_\varepsilon\sqrt{\sum_{j=1}^{d} T_0\log(1/\delta_j)}$$

$$= L\sigma_\varepsilon\sqrt{dT_0\log(d/\delta)},$$

with probability at least $1 - \delta$, where the last line follows from a union bound where $\delta_j = \delta/d$ for all $j \in [d]$.

**Bounding** $\lambda_{min}\left(\sum_{t=1}^{T_0/2} \mathbf{x}_t\mathbf{x}_t^\top\right)$

Let $Y = \sum_{t=1}^{T_0/2} \mathbf{x}_t\mathbf{x}_t^\top$.

$$\mu_{min} = \lambda_{min}(\mathbb{E}[Y])$$

$$= \lambda_{min}(\mathbb{E}[\sum_{t=1}^{T_0/2} \mathbf{x}_t\mathbf{x}_t^\top])$$

$$= \lambda_{min}(\frac{1}{2}T_0\mathbb{E}[\mathbf{x}_1\mathbf{x}_1^\top])$$

$$= \frac{1}{2}T_0\lambda_{min}(\mathbb{E}[\mathbf{x}_1\mathbf{x}_1^\top])$$

$$= \frac{1}{2}T_0(\lambda_{min}(\sigma_x^2\mathbb{I}_{d\times d} + \mathbb{E}[\mathbf{x}_1]\mathbb{E}[\mathbf{x}_1^\top]))$$

$\lambda_{min}(\sigma_x^2\mathbb{I}_{d\times d})$ and $\lambda_{min}(\mathbb{E}[\mathbf{x}_1]\mathbb{E}[\mathbf{x}_1^\top])$ commute, so

$$\mu_{min} = \frac{1}{2}T_0(\lambda_{min}(\sigma_x^2\mathbb{I}_{d\times d}) + \lambda_{min}(\mathbb{E}[\mathbf{x}_1]\mathbb{E}[\mathbf{x}_1^\top]))$$

$$= \frac{1}{2}T_0\lambda_{min}(\sigma_x^2\mathbb{I}_{d\times d})$$

$$= \frac{1}{2}T_0\sigma_x^2\lambda_{min}(\mathbb{I}_{d\times d})$$

$$= \frac{1}{2}T_0\sigma_x^2$$

We use the following result to bound $\lambda_{min}\left(\sum_{t=1}^{T_0/2} \mathbf{x}_t\mathbf{x}_t^\top\right)$ with high probability:

**Theorem A.4** (Matrix Chernoff). *Consider a finite sequence $\{X_t\}_{t=1}^T$ of independent, random, Hermitian matrices with common dimension $d$. Assume that*

$$0 \leq \lambda_{min}(X_t) \text{ and } \lambda_{max}(X_t) \leq L' \text{ for each index } t$$

*Introduce the random matrix*

$$Y = \sum_{t=1}^T X_t.$$

*Define the minimum eigenvalue $\mu_{min}$ of the expectation $\mathbb{E}[Y]$:*

$$\mu_{min} = \lambda_{min}(\mathbb{E}[Y]) = \lambda_{min}\left(\sum_{t=1}^T \mathbb{E}[X_t]\right)$$

*Then,*

$$\mathbb{P}(\lambda_{min}(Y) \leq (1-\varepsilon)\mu_{min}) \leq d\left(\frac{e^{-\varepsilon}}{(1-\varepsilon)^{1-\varepsilon}}\right)^{\mu_{min}/L'}$$

*for $\varepsilon \in [0, 1)$.*

$\lambda_{max}(\mathbf{x}_t\mathbf{x}_t^\top) \leq dL$, so $L' = dL$. Pick $\varepsilon = \frac{1}{2}$. Applying Theorem A.4 to $\lambda_{min}(\sum_{t=1}^{T_0/2} \mathbf{x}_t\mathbf{x}_t^\top)$,

$$\mathbb{P}(\lambda_{min}(\sum_{t=1}^{T_0/2} \mathbf{x}_t\mathbf{x}_t^\top) \leq \frac{1}{4}T_0\sigma_x^2) \leq d\left(\frac{1}{2}e\right)^{-\frac{T_0\sigma_x^2}{4dL}} \tag{1}$$

After inverting the bound, we see that $\lambda_{min}(\sum_{t=1}^{T_0/2} \mathbf{x}_t\mathbf{x}_t^\top) \geq \frac{1}{4}T_0\sigma_x^2$ with probability at least $1 - \delta$ if $T_0 \geq \frac{4dL}{\sigma_x^2 \log \frac{1}{2}e} \log \frac{d}{\delta}$.

Combining the bounds for $\left\|\sum_{t=1}^{T_0/2} \mathbf{x}_t\varepsilon_t\right\|_2$ and $\lambda_{min}\left(\sum_{t=1}^{T_0/2} \mathbf{x}_t\mathbf{x}_t^\top\right)$ via a union bound, we obtain our stated result.

$\square$

**Theorem A.5.** *If $T_0 \geq \frac{4dL}{\sigma_x^2 \log \frac{1}{2}e} \log(2dT^4)$, Algorithm `Explore-First` achieves expected strategic regret*

$$\mathbb{E}[R(T)] \leq T_0 + T\frac{16L\sigma_\varepsilon\sqrt{d\log(4dT^4)}}{\sqrt{T_0}\sigma_x^2}$$

*Proof.*

$$\mathbb{E}[R(T)] = \sum_{t=1}^T \mathbb{E}[r(\pi_S(\mathbf{x}_t))|\mathbf{x}_t] - \sum_{t=1}^T \mathbb{E}[r(\pi_t(\mathbf{x}_t'))|\mathbf{x}_t']$$

$$\leq T_0 + \sum_{t=T_0+1}^T \mathbb{E}[r(\pi_S(\mathbf{x}_t))|\mathbf{x}_t] - \sum_{t=1}^T \mathbb{E}[r(\pi_t(\mathbf{x}_t'))|\mathbf{x}_t']$$

$$= T_0 + \sum_{t=T_0+1}^T \langle\boldsymbol{\theta}_{a_t^*} - \boldsymbol{\theta}_{a_t}, \mathbf{x}_t\rangle$$

If $T_0 \geq \frac{8dL}{\sigma_x^2 \log \frac{1}{2}e} \log \frac{2d}{\delta}$, the following two events hold simultaneously with probability at least $1 - \delta$:

$$|\langle\hat{\boldsymbol{\theta}}_1 - \boldsymbol{\theta}_1, \mathbf{x}\rangle| \leq \frac{4L\sigma_\varepsilon\sqrt{d\log(4d/\delta)}}{\sqrt{T_0}\sigma_x^2}$$

$$|\langle\hat{\boldsymbol{\theta}}_2 - \boldsymbol{\theta}_2, \mathbf{x}\rangle| \leq \frac{4L\sigma_\varepsilon\sqrt{d\log(4d/\delta)}}{\sqrt{T_0}\sigma_x^2}$$

If $a_t$ is the suboptimal action to assign to agent $t$, then with probability at least $1 - \delta$,

$$
\langle \boldsymbol{\theta}_{a_t}, \mathbf{x}_t \rangle + \frac{4L\sigma_\varepsilon \sqrt{d \log(4d/\delta)}}{\sqrt{T_0}\sigma_x^2} \geq \left\langle \hat{\boldsymbol{\theta}}_{a_t}, \mathbf{x}_t \right\rangle \geq \left\langle \hat{\boldsymbol{\theta}}_{a_t^*}, \mathbf{x}_t \right\rangle \tag{2}
$$
$$
\geq \left\langle \boldsymbol{\theta}_{a_t^*}, \mathbf{x}_t \right\rangle - \frac{4L\sigma_\varepsilon \sqrt{d \log(4d/\delta)}}{\sqrt{T_0}\sigma_x^2}
$$

Therefore, $\left\langle \boldsymbol{\theta}_{a_t^*} - \boldsymbol{\theta}_{a_t}, \mathbf{x}_t \right\rangle \leq \frac{8L\sigma_\varepsilon \sqrt{d \log(4d/\delta)}}{\sqrt{T_0}\sigma_x^2}$ with probability at least $1 - \delta$.

By an argument similar to the proof of Proposition 4.2, the `Explore-First`'s strategic regret is the same as that of making decisions according to $\hat{\boldsymbol{\beta}}_{12}$ on non-strategic agents. Using this observation, we can set $\delta = \frac{1}{T^4}$ and upper-bound expected strategic regret by

$$
\mathbb{E}[R(T)] \leq T_0 + T\frac{16L\sigma_\varepsilon \sqrt{d \log(4dT^4)}}{\sqrt{T_0}\sigma_x^2}
$$

$\square$

**Corollary A.6.** *If* $T_0 = \frac{4L^{2/3}\sigma_\varepsilon^{2/3}(d \log(4dT^4))^{1/3}}{\sigma_x^{4/3}}T^{2/3}$ *and* $T \geq \frac{2L^{1/2}d}{\sigma_\varepsilon \sigma_x (\log \frac{1}{2}e)^{3/2}} \log \frac{16L^{1/2}d^2}{\sigma_\varepsilon \sigma_x (\log \frac{1}{2}e)^{3/2}}$, *then*

$$
\mathbb{E}[R(T)] \leq \frac{16L^{2/3}\sigma_\varepsilon^{2/3}(d \log(4dT^4))^{1/3}}{\sigma_x^{4/3}}T^{2/3}
$$

The proof of Corollary A.6 follows straightforwardly from Theorem A.5 and the inequality $\log x \leq \alpha x - \log \alpha - 1$ for any $\alpha, x > 0$.

# B Proofs from Section 5

## B.1 Proof of Lemma 5.2

**Lemma B.1.** *To prevent gaming, the decision maker must assign action $a_2$ only if* $\langle \boldsymbol{\beta}_{12}, \mathbf{x}_t \rangle + \varepsilon_{12} < 0$ *and* $\langle \boldsymbol{\beta}_{23}, \mathbf{x}_t \rangle - \varepsilon_{23} \geq 0$, *where* $\varepsilon_{12} \geq \delta\|\boldsymbol{\beta}_{12}\|_2 - \alpha_2$ *and* $\varepsilon_{23} > \delta\|\boldsymbol{\beta}_{23}\|_2 - \alpha_2$.

*Proof.* Since $r^A(1) = r^A(3)$, agents are indifferent between receiving actions 1 and 3, and so $\boldsymbol{\beta}_{13}$ is the optimal decision boundary between the $a = 1$ and $a = 3$ equivalence regions. Due to the structure of the distribution over agents, it is w.l.o.g. to consider linear decision boundaries which are parallel to $\boldsymbol{\beta}_{12}$ and $\boldsymbol{\beta}_{23}$.

Since $r^A(2) > r^A(1)$, an agent with private type $\mathbf{x}_t \in \mathcal{X}_2$ has incentive to choose their context such that $\langle \boldsymbol{\beta}_{12}, \mathbf{x}_t' \rangle + \varepsilon_{12} < 0$, where $\mathbf{x}_t'$ is agent $t$'s context. It is w.l.o.g. to assume the agent modifies their private type in the direction perpendicular to $\boldsymbol{\beta}_{12}$. The most the agent can modify in this direction is $\frac{-\delta\boldsymbol{\beta}_{12}}{\|\boldsymbol{\beta}_{12}\|_2}$. Therefore if we want $\langle \boldsymbol{\beta}_{12}, \mathbf{x}_t' \rangle + \varepsilon_{12} \geq 0$,

$$
\langle \boldsymbol{\beta}_{12}, \mathbf{x}_t - \frac{\delta\boldsymbol{\beta}_{12}}{\|\boldsymbol{\beta}_{12}\|_2} \rangle + \varepsilon_{12} \geq 0
$$
$$
\langle \boldsymbol{\beta}_{12}, \mathbf{x}_t \rangle - \delta\|\boldsymbol{\beta}_{12}\|_2 + \varepsilon_{12} \geq 0
$$
$$
\alpha_2 - \delta\|\boldsymbol{\beta}_{12}\|_2 + \varepsilon_{12} \geq 0
$$

Similarly for an agent $\mathbf{x}_t \in \mathcal{X}_3$,

$$
\langle \boldsymbol{\beta}_{23}, \mathbf{x}_t + \frac{\delta\boldsymbol{\beta}_{23}}{\|\boldsymbol{\beta}_{23}\|_2} \rangle - \varepsilon_{23} < 0
$$
$$
\langle \boldsymbol{\beta}_{23}, \mathbf{x}_t \rangle + \delta\|\boldsymbol{\beta}_{23}\|_2 - \varepsilon_{23} < 0
$$
$$
-\alpha_2 + \delta\|\boldsymbol{\beta}_{23}\|_2 - \varepsilon_{23} < 0
$$

$\square$

## B.2 Proof of Lemma 5.3

**Lemma B.2.** *To correctly classify agents with $\mathbf{x}_t = \mathbf{x}_1$, the decision maker must assign action $a_2$ only if $\langle \boldsymbol{\beta}_{12}, \mathbf{x}_t \rangle + \varepsilon_{12} < 0$ and $\langle \boldsymbol{\beta}_{23}, \mathbf{x}_t \rangle - \varepsilon_{23} \geq 0$, where $\varepsilon_{12} \leq -\boldsymbol{\beta}_{12}[2](\alpha_1 + \delta)$ and $\varepsilon_{23} \leq \boldsymbol{\beta}_{23}[2](\alpha_1 + \delta)$.*

*Proof.* It is w.l.o.g. to assume that $\mathbf{x}'_t = [0 \quad \alpha_1 + \delta]^\top$. If we want $\langle \boldsymbol{\beta}_{12}, \mathbf{x}'_t \rangle + \varepsilon_{12} < 0$, then

$$\boldsymbol{\beta}_{12}[2](\alpha_1 + \delta) + \varepsilon_{12} < 0$$

Similarly, if we want $\langle \boldsymbol{\beta}_{23}, \mathbf{x}'_t \rangle - \varepsilon_{23} \geq 0$, then

$$\boldsymbol{\beta}_{23}[2](\alpha_1 + \delta) - \varepsilon_{23} \geq 0$$

$\square$

