# OpenReview forum: "Strategy-Aware Contextual Bandits"
_NeurIPS.cc/2022/Workshop/TSRML — TSRML2022_

### Official Review · Reviewer_uMkr · 2022-10-19

**Overall Rating:** 6

**Summary:**

* This theoretical work introduces a model of contextual bandits with strategic manipulations, provides a vanishing-strategic-regret algorithm for $n=2$ actions, and impossibility results $n\ge 3$ actions.
* In the proposed model, a decision maker interacts with a sequence of strategic agents. Each agent is associated with a feature vector $\mathbf{x}$, and may strategically manipulate its reported vector $\mathbf{x}\mapsto\mathbf{x}’$ to get a better reward. The decision maker’s action space $\mathcal{A}$ is assumed to be discrete, their reward for each action is assumed to be linear in the reported context $\mathbf{x}’$, and the agent’s reward is assumed to be a function of the chosen action alone. Strategic manipulation $\mathbf{x}\mapsto\mathbf{x}’$ assumed to be achieved through reward maximization under a limited $l_2$-norm budget.
* For the case of $n=2$ actions, authors present a vanishing-strategic-regret algorithm. This is achieved using an explore-then-exploit approach, relying on the fact that stateless agents don’t have an incentive to modify their actions when the algorithm action is constant.
* For the case of $n\ge 3$ actions, authors present cases where no linear policy can achieve perfect classification under strategic manipulation, leading to a linear regret lower bound.


**Strengths:**

* Presentation is very clear and coherent.
* Theoretical results are interesting and seem to be non-trivial.


**Weaknesses:**

* Relation to the “Responsible and Trustworthy Machine Learning” theme could be further explored, and I feel it would be interesting to discuss connections to real-world settings. For example:
  * When do we expect model assumptions to be valid? (e.g linear agent reward, independent agents, $l_2$-bounded strategic manipulation).
  * What would be the biggest benefit from applying the presented algorithms (or similar ones) in practice?
* In the current formulation of the model, agents are assumed to be “stateless”, myopically interacting with the decision maker for a single round. In contrast, agents in real-world settings (e.g marketing firms) may also be interested in long-term benefits, and may strategically manipulate the decision maker in the early exploration stages to achieve better reward in the exploitation stage. It is not clear whether Algorithm 1 is robust to such behavior. For example, what happens if the agent is also assumed to act according to a no-regret policy?
* No empirical evaluation of results.


**Overall Recommendation:**

Despite the gaps in its core behavioral assumptions and the lack of details about practical applicability, this work does provide meaningful contributions to our theoretical understanding of strategic classification. As a result, I rate it as "Marginally above acceptance threshold".

**Review Confidence:**

3: The reviewer is fairly confident that the evaluation is correct

---

### Decision · Program_Chairs · 2022-10-23

**Decision:**

Accept

**Comment:**

Following the recommendation from the review, the submission is accepted.

Please follow the review to add more discussion about: When do we expect model assumptions to be valid? How and what may be the pros and cons if applying the proposed algorithms in practice?